# Effects of Different Fiber Substrates on In Vitro Rumen Fermentation Characteristics and Rumen Microbial Community in Korean Native Goats and Hanwoo Steers

**Seon-Ho Kim [1] and Ha-Guyn Sung [2,*]**

[1] Ruminant Nutrition and Anaerobe Laboratory, Department of Animal Science and Technology, Sunchon National University, 413 Jungangro, Suncheon 57922, Jeonnam, Korea
[2] Animal Feeding and Environment Laboratory, Department of Animal Science, Sangji University, 83 Sangdae-gil, Wonju-si 26339, Gangwon-do, Korea
\* Correspondence: haguyn@hanmail.net; Tel.: +82-33-730-0536 or +82-10-4045-2045

**Abstract:** Korean native goats (*Capra hircus coreanae*) (KNG) and Hanwoo (*Bos taurus coreanae*) are indigenous breeds inhabiting Korea. This study compared the in vitro rumen fermentation characteristics, dry matter (DM) degradation, and ruminal microbial communities of Korean native goats and Hanwoo steers consuming rice hay (RH) and cotton fiber (CF). The pH, ammonia-nitrogen ($NH_3$-N), and total volatile fatty acids (VFAs) production significantly differ ($p < 0.05$) across species in all incubation times. After 24 h, the pH, $NH_3$-N, and total VFAs production were higher in Korean native goats than in Hanwoo steers. Total gas, molar proportion of propionate, and total VFAs were higher ($p < 0.05$) in RH than in CF for both ruminant species. DM digestibility of both substrates were higher ($p < 0.05$) in Hanwoo steers than in KNG. Both treatments in KNG produced higher ($p < 0.01$) microbial DNA copies of general bacteria than those in Hanwoo steers. *Butyrivibrio fibrisolvens* and *Fibrobacter succinogenes* had significantly higher DNA copies under RH and CF in Hanwoo steers than in Korean native goats. *B. fibrisolvens*, *Ruminococcus albus*, and *Ruminococcus flavifaciens* after 24 h of incubation had a higher abundance ($p < 0.05$) in RH than in CF. Overall results suggested that rumen bacteria had host-specific and substrate-specific action for fiber digestion and contribute to improving ruminal functions of forage utilization between ruminant species.

**Keywords:** rumen fermentation characteristics; DM degradation; rumen microbial community; Korean native goats; Hanwoo steers

## 1. Introduction

Feeding systems in the modern ruminant industry are progressively shifting to diets containing a relatively high proportion of concentrate to satisfy the increasing energy requirements of intensively managed animals [1]. However, dietary fiber, which is related to adequate salivation, optimal pH for cellulolytic micro-organisms, and energy supply, is an essential nutrient in feed and is useful for the maintenance of normal rumen function [1–5]. Useful dietary fibers are crop residues, such as rice hay and cotton, which are commonly used as feedstuffs for ruminant nutrition [1,6]. Rice hay, a common cereal crop by-product in Asian countries, is an important feed source for ruminants. Studies have revealed that increasing the proportion of rice straw increases dietary NDF but decreases the utilization of nutrients [1]. Cotton by-products, such as cottonseed hulls, cotton gin trash, and textile mill waste, are primarily roughage or fiber sources, though cottonseed meal is primarily a protein and phosphorus source [6]. Whole cottonseed is a by-product of cotton ginning, and is a relatively high source of energy, fiber, and protein in rations fed to ruminants [7,8]; its incorporation into ruminant diets is considered beneficial because of its favorable nutritional value [8]. Arieli [9] and Holter et al. [10] stated that using whole cottonseed in

ruminant diets is advantageous, considering its low digestive heat increment, which might decrease metabolic heat production in cattle, particularly at high ambient temperatures.

Conversely, other in vivo studies have indicated that the inclusion of whole cottonseed in the diets of steers or sheep reduced fiber digestibility, dry matter, and relative abundance of rumen microbes [11]. Palmquist [12] stated that the long cotton linters retained on the hulls delayed digestibility when incubated for periods varying from 12 to 140 h on in sacco and in vitro. This might be due to the long cotton fibers being composed of highly crystalline structures that hydrate slowly [13]. The long cotton fiber digestion lag could increase the pool size of undigested fiber in whole linted cottonseed-fed ruminants [12]. Considering the different dietary fibers as a source of energy and essential nutrients, ruminant species, such as cattle and goats, may have different feeding behaviors and utilization, and levels and rates of intake [1,14,15]. In connection to this, no research has yet been published concerning the digestibility of cotton lint and rice hay using in vitro ruminal fermentation with rumen fluid of cattle and goats. Therefore, in the present study, ruminal fermentation characteristics, DM degradation, and rumen microbial community analyses were conducted using ruminal fluid from Korean native goats and Hanwoo steers to evaluate and compare the effects of rice hay (RH) and cotton fiber (CF) as substrates in in vitro trials.

## 2. Materials and Methods

### 2.1. Animal Care

All experimental procedures were performed in accordance with the Animal Experimental Guidelines of the Sunchon National University Institutional Animal Care and Use Committee (SCNU-IACUC), Republic of Korea. The experimental protocols were reviewed and approved by SCNU-IACUC (approval number: SCNU IACUC-2018-01).

### 2.2. Animals, Rumen Fluid Collection, In Vitro Rumen Fermentation, and DM Degradation

Four Hanwoo steers ($550 \pm 47$ kg body weight; 20 months of age) and four Korean native black goats ($73 \pm 12$ kg body weight; 15 months of age) were used to provide rumen fluid for in vitro rumen fermentation. Animals were housed in the experimental farm at Sunchon National University. Ruminal fluid donors were fed twice daily at 09:00 and 18:00 with a mixture of concentrate and Italian ryegrass hay (4:6 ratio) for more than 2 weeks. Rumen contents were collected before the morning feeding and were obtained through stomach intubation. The ruminal contents of both donors were separately filtered through four layers of surgical gauze and placed in a stainless-steel vacuum bottle. After collection, the filtrates were sealed, maintained at 39 °C, and immediately transported to the laboratory [16]. The buffer medium used in this study consisted of (per liter) 0.45 g dipotassium phosphate ($K_2HPO_4$), 0.45 g potassium dihydrogen phosphate ($KH_2PO_4$), 0.9 g ammonium sulfate (($NH_4)_2SO_4$), 0.12 g calcium chloride dihydrate ($CaCl_2 \cdot 2H_2O$), 0.9 g sodium chloride (NaCl), 0.19 g magnesium sulfate heptahydrate ($MgSO_4 \cdot 7H_2O$), 1.0 g trypticase peptone, 1.0 g yeast extract, and 0.6 g L-cysteine hydrochloride [17]. The prepared buffer was autoclaved for 15 min at 121 °C, maintained in a 39 °C water bath, and flushed with $CO_2$ gas while adjusting its pH to 6.9 using 10 N NaOH [18]. The rumen fluid was pooled and strained again using a four-layer surgical gauze placed in a funnel, and the filtrate was allowed to flow into a 1 L graduated cylinder with continuous bubbling of $CO_2$ gas. The filtered rumen fluid was then mixed with the prepared buffer medium at a ratio of 1:3 (*v/v*), under a constant flow of $CO_2$ gas [19]. The pH was adjusted again to 6.9, before filling the serum bottle with buffered rumen fluid. Rice hay (RH) and cotton fiber (CF) served as treatment substrates and were ground to pass through a 2 mm sieve. The chemical composition of the fiber substrates was analyzed according to AOAC method [20] and its composition is shown in Table 1. Then, 1 g of the weighed substrate from each treatment was added to the serum bottles. One hundred milliliters of buffered rumen fluid were filled into individual serum bottles containing the substrate under a constant flow of $CO_2$ gas. The bottles were subsequently sealed with rubber septum stoppers and

aluminum caps and incubated at 39 °C for 6, 12, and 24 h with spontaneous shaking at 100 rpm [21]. Three replicates each was prepared for all treatments at different incubation times (6, 12, and 24 h).

**Table 1.** Chemical composition of fiber substrates. All components except moisture content are expressed on a percentage of dry matter basis.

| Chemical Composition (%) | RH | CF |
|---|---|---|
| Moisture | 7.76 | 4.39 |
| Crude protein | 2.74 | 1.44 |
| Ether extract | 1.64 | 0.39 |
| Crude fiber | 40.80 | 84.88 |
| Crude ash | 8.59 | 1.50 |
| Acid detergent fiber | 44.98 | 85.33 |
| Neutral detergent fiber | 75.68 | 87.91 |

RH, rice hay; CF, cotton fiber.

For DM degradation, samples from each treatment were dried in an oven for 48 h at 80 °C and ground to pass through a 1 mm screen [22]. Samples weighing 1 g (DM basis) were placed in nylon bags (5 × 10 cm; 45 µm pore size), and the bag openings were tied with nylon strings. The nylon bags were then placed in different serum bottles. The buffered rumen fluid was prepared similarly with the buffer used from the in vitro gas production system. The buffered rumen fluid was placed in a warm water bath (39–40 °C) under a constant flow of $CO_2$ gas for 30 min. Then, the buffered rumen fluid was poured into individual serum bottles containing the bags under a constant flow of $CO_2$ gas. The incubation times were 6 h, 12 h, and 24 h, with three replicates per treatment. After each incubation, the nylon bag was removed from the serum bottle, immersed in clear water, and washed under running water. The samples for each incubation time were dried at 80 °C for 24 h in a dry oven immediately after washing.

### 2.3. Analysis of In Vitro Rumen Fermentation and DM Degradation

A press-and-sensor machine (EA-6; Laurel Electronics, Inc., Costa Mesa, CA, USA) was used to measure the total gas (TG). The pH was determined using Schott® Instruments Lab 860 (SI Analytics GmbH; D-55122 Mainz Deutschland, Germany, Allemagne) after each serum bottle was opened. Rumen fermenta from each bottle was collected and placed in 1.5 mL microcentrifuge tubes and stored at −80 °C. These samples were later thawed at room temperature and centrifuged at 13,000× $g$ for 15 min at 4 °C using a Micro 17TR centrifuge (Hanil Science Industrial, Incheon, Korea). The supernatant was then used for $NH_3$-N and VFA analyses [23]. Using a Libra S22 spectrophotometer (Biochrom Ltd., Cambridge CB4 0FJ, England) at an absorbance of 630 nm, the $NH_3$-N concentration was measured according to the methods described by Chaney and Marbach [24]. Volatile fatty acids were analyzed using high performance liquid chromatography (Agilent Technologies 1200 series, Tokyo, Japan) with a UV detector (210 nm and 220 nm) and a Metacarb87H (Agilent Technologies, Minnetonka, MN, USA) column using 0.0085N $H_2SO_4$ as a buffer at a flow rate of 0.6 mL/min and temperature column of 35 °C. The DM degradation was calculated according to the protocol described by Van Emon et al. [25].

### 2.4. Quantification of Microbial DNA Copies from In Vitro Rumen Fermentation

Microcentrifuge tubes containing the rumen fluid were centrifuged at 17,000× $g$ for 15 min at 4 °C. The supernatant was discarded, and the isolated pellets were used to extract microbial DNA using a FastDNA SPIN Kit (MP Biomedicals, Solon, OH, USA), following the manufacturer's protocol. The DNA was resuspended in 50 µL of DES (DNase/pyrogen-free water). The quality and quantity of the DNA were assessed using an Optizen NanoQ spectrophotometer (Optizen, Korea) and agarose gel electrophoresis. The DNA samples were stored at -20 °C until subsequent analysis [23].

The microbial targets as well as the primer sequences for the real-time PCR assays used in the present study are summarized in Table 2. Quantitative real-time PCR (qPCR) was performed using an Eco RealTime PCR (Illumina, San Diego, CA, USA) in a 20 µL reaction mixture consisting of 10 µL of 2× QuantiSpeed SYBR No-Rox mix (PhileKorea, Daejeon, Korea), 0.8 µL each of 10 pmol primers, and 50 ng of template DNA. The qPCR reactions were performed under thermal cycler conditions of one cycle at 50 °C for 2 min and 95 °C for 2 min, followed by 40 cycles at 95 °C for 15 s, 60 °C for 1 min, and 72 °C for 30 s. Amplification of samples, standards, and the negative control (without the DNA template) were run in triplicate. The standard curves were generated using 10-fold serial dilutions of each standard DNA sample containing the target gene sequences of the respective microbial groups. The relative abundance of each microbial population was expressed as DNA copies of the target gene per 50 ng of genomic DNA (gDNA) in rumen fluid [23].

**Table 2.** Primer information used for real-time PCR.

| Target Micro-Organisms | Primer Sequence (5 → 3′) | References |
|---|---|---|
| General bacteria | Forward: CGGCAACGAGCGCAACCC<br>Reverse: CCATTGTAGCACGTGTGTAGCC | [26] |
| Total fungi | Forward: GAGGAAGTAAAAGTCGTAACAAGGTTTC<br>Reverse: CAAATTCACAAAGGGTAGGATGATT | [26] |
| Total protozoa | Forward: GCTTTCGWTGGTAGTGTATT<br>Reverse: CTTGCCCTCYAATCGTWCT | [27] |
| *Ruminococcus albus* | Forward: CCCTAAAAGCAGTCTTAGTTCG<br>Reverse: CCTCCTTGCGGTTAGAACA | [28] |
| *Ruminococcus flavefaciens* | Forward: CGAACGGAGATAATTTGAGTTTACTTAGG<br>Reverse: CGGTCTCTGTATGTTATGAGGTATTACC | [26] |
| *Fibrobacter succinogenes* | Forward: GTTCGGAATTACTGGGCGTAAA<br>Reverse: CGCCTGCCCCTGAACTATC | [26] |
| *Butyrivibrio fibrisolvens* | Forward: TAACATGAGAGTTTGATCCTGGCTC<br>Reverse: CGTTACTCACCCGTCCGC | [26] |

*2.5. Statistical Analysis*

All data obtained in this experiment were collected for each incubation period, and statistical analysis was performed using the standard Statistical Analysis System (SAS) (version 9.4; SAS Institute Inc., Cary, NC, USA). The data were subjected to ANOVA using the mixed procedure of SAS. The model included the fixed effects animal species, treatments, and an interaction term of animal species and treatments, according to the following statistical model:

$$Y_{ijk} = \mu + \alpha_i + \beta_j + (\alpha\beta)_{ij} + \varepsilon_{ijk},$$

where $Y_{ijk}$ = response variable, $\mu$ = overall mean, $\alpha_i$ = fixed effect of the ith animal species (S), $\beta_j$ = fixed effect of the jth treatment (T), $(\alpha\beta)_{ij}$ = interaction between species and treatment (S × T), and $\varepsilon_{ijk}$ = random error. A difference of $p < 0.05$ was considered significant.

**3. Results**

*3.1. Effect of Rice Hay and Cotton on In Vitro Rumen Fermentation Parameters*

The in vitro rumen fermentation parameters of Korean native goats and Hanwoo steers are presented in Table 3. There was a significance in total gas production between species at 12 and 24 h of incubation, while it is significant in between treatments in all incubation periods. The total gas production in Hanwoo steers was higher ($p < 0.05$) than in KNG. Moreover, total gas production was higher ($p < 0.05$) in RH compared with CF in both species. No significant interaction between species and treatments was observed

in total gas production in all incubation times. There were significant differences between the species in pH and $NH_3$-N in all incubation periods. Rumen pH was higher ($p > 0.05$) in KNG than in Hanwoo steers. However, pH was higher in CF than in RH treatment at 12 h ($p = 0.020$). Meanwhile, a significant effect on the interaction between species and treatments was observed only at 6 h of incubation ($p = 0.003$). In terms of $NH_3$-N production, higher ($p < 0.05$) $NH_3$-N was observed in KNG than in Hanwoo steers in all incubation periods. However, there was no significance between treatments and interaction between species and treatments ($p < 0.05$) in all incubations. On the other hand, molar proportions of propionate and butyrate was significantly different between species at 12 and 24 h of incubation. The molar proportion of propionate and butyrate was significantly higher in KNG and Hanwoo steers, respectively. In addition, molar proportions of acetate and butyrate were significantly different across treatments at 12 and 24 h. Acetate and butyrate were significantly higher ($p < 0.05$) in CF than in RH treatment in both species. A significant interaction between species and treatments ($p < 0.05$) were observed in propionate in all incubation period, while in butyrate at 12 and 24 h. Total VFA was significantly different between species in all incubation periods, while there was significance ($p < 0.05$) across treatments at 6 and 12 h. Total VFA was higher ($p < 0.05$) in KNG than in Hanwoo steers in all incubation periods. Meanwhile, the interaction effect ($p < 0.05$) between species and treatments was found at 12 h only.

**Table 3.** Effect of rice hay and cotton on in vitro ruminal fermentation parameters after 24 h incubation.

| Parameters | | Treatment [1] | | | | SEM [2] | p Value [3] | | |
| --- | --- | --- | --- | --- | --- | --- | --- | --- | --- |
| | | Korean Native Goats | | Hanwoo Steers | | | | | |
| | Time (h) | RH | CF | RH | CF | | S | T | S × T |
| Total gas (mL) | 6 | 6.67 | 3.67 | 5.33 | 1.33 | 0.938 | 0.108 | 0.009 | 0.635 |
| | 12 | 18.33 | 12.67 | 12.33 | 6.33 | 0.554 | <0.001 | <0.001 | 0.789 |
| | 24 | 22.67 | 13.67 | 33.67 | 28.00 | 0.721 | <0.001 | <0.001 | 0.061 |
| pH | 6 | 6.35 [c] | 6.41 [b] | 6.51 [a] | 6.41 [b] | 0.015 | 0.002 | 0.271 | 0.003 |
| | 12 | 6.47 | 6.51 | 6.31 | 6.37 | 0.014 | <0.001 | 0.020 | 0.479 |
| | 24 | 6.37 | 6.41 | 6.20 | 6.23 | 0.014 | <0.001 | 0.106 | 0.815 |
| $NH_3$-N (mg/dL) [4] | 6 | 11.9 [a] | 10.05 | 7.34 | 7.18 | 0.585 | <0.001 | 0.148 | 0.213 |
| | 12 | 15.55 | 15.11 | 9.51 | 11.04 | 0.962 | 0.001 | 0.595 | 0.350 |
| | 24 | 14.64 | 16.29 | 11.23 | 10.88 | 1.282 | 0.013 | 0.655 | 0.491 |
| Acetate (mol/100 mol) | 6 | 61.24 | 59.64 | 61.15 | 62.03 | 0.588 | 0.115 | 0.594 | 0.094 |
| | 12 | 62.43 | 60.63 | 61.38 | 60.23 | 0.340 | 0.143 | 0.011 | 0.491 |
| | 24 | 57.99 | 60.70 | 57.45 | 60.47 | 0.436 | 0.460 | 0.001 | 0.758 |
| Propionate (mol/100 mol) | 6 | 22.94 [ab] | 23.66 [ab] | 24.26 [a] | 22.52 [b] | 0.310 | 0.882 | 0.233 | 0.014 |
| | 12 | 23.36 [c] | 24.82 [b] | 26.46 [a] | 26.18 [a] | 0.239 | 0.001 | 0.131 | 0.038 |
| | 24 | 28.87 [b] | 26.36 [c] | 31.97 [a] | 27.21 [bc] | 0.506 | 0.007 | 0.001 | 0.007 |
| Butyrate (mol/100 mol) | 6 | 15.83 | 16.70 | 14.58 | 15.45 | 0.700 | 0.163 | 0.317 | 0.994 |
| | 12 | 14.22 [ab] | 14.55 [a] | 12.16 [c] | 13.59 [b] | 0.166 | 0.025 | 0.002 | <0.001 |
| | 24 | 13.14 [a] | 12.95 [a] | 10.59 [b] | 12.33 [a] | 0.325 | 0.003 | 0.070 | 0.031 |
| A/P ratio | 6 | 2.52 [b] | 2.67 [a] | 2.76 [a] | 2.52 [b] | 0.040 | 0.404 | 0.411 | 0.005 |
| | 12 | 2.44 | 2.68 | 2.30 | 2.32 | 0.038 | 0.003 | 0.063 | 0.102 |
| | 24 | 2.30 | 2.01 | 2.22 | 1.80 | 0.047 | 0.019 | <0.001 | 0.224 |
| Total VFAs (mmol/L) | 6 | 34.62 | 32.79 | 25.75 | 25.02 | 0.411 | <0.001 | 0.018 | 0.240 |
| | 12 | 40.44 [a] | 36.55 [b] | 31.40 [c] | 29.68 [c] | 0.259 | <0.001 | <0.001 | 0.007 |
| | 24 | 44.67 | 42.43 | 40.04 | 38.47 | 1.189 | 0.007 | 0.151 | 0.785 |

[1] Treatment: RH, rice hay; CF, cotton fiber; [2] SEM, standard error of the mean; [3] S, species; T, treatment; S × T: interaction between species and treatment; [4] $NH_3$-N, ammonia nitrogen; A/P, acetate to propionate ratio; [a–c] Means with different superscripts in a row differ significantly ($p < 0.05$).

### 3.2. In Vitro DM Degradation of Rice Hay and Cotton

There were significant differences in in vitro DM degradation between the treatments at all incubation times (Table 4). In vitro DM degradation was higher ($p < 0.01$) in RH than in CF in all incubation times. There were significant differences in in vitro DM

degradation between species at 12 and 24 h of incubation. The in vitro DM degradation of the substrates was higher ($p < 0.05$) in Hanwoo steers than in Korean native goats. This might be explained by the higher copy number of microbial DNA of cellulolytic bacteria, such as *Butyrivibrio fibrisolvens* and *Fibrobacter succinogenes*, in the rumen fluid of Hanwoo steers at 12 and 24 h incubation (Table 5). In vitro DM digestibility was higher ($p < 0.05$) in RH than CF for both ruminant species throughout the incubation period. Furthermore, there was evidence of interaction ($p = 0.027$) between the species and treatments at 24 h of incubation.

**Table 4.** In vitro DM degradation of rice hay and cotton fiber in Korean native goats and Hanwoo steers.

| Parameter | Time (h) | Treatment [1] | | | | SEM [2] | p Value [3] | | |
| | | Korean Native Goats | | Hanwoo Steers | | | | | |
| | | RH | CF | RH | CF | | S | T | S × T |
|---|---|---|---|---|---|---|---|---|---|
| DM degradation (%) | 6 | 5.81 | 1.32 | 5.92 | 1.18 | 0.691 | 0.456 | 0.001 | 0.545 |
| | 12 | 9.34 | 2.53 | 9.03 | 3.29 | 0.483 | 0.068 | <0.001 | 0.188 |
| | 24 | 16.57 [a] | 7.36 [b] | 19.47 [a] | 15.93 [b] | 0.964 | 0.001 | <0.001 | 0.027 |

[1] Treatment: RH, rice hay; CF, cotton fiber; [2] SEM, standard error of the mean; [3] S, species; T, treatment; S × T: interaction between species and treatment. [a,b] Means with different superscripts in a row differ significantly ($p < 0.05$).

**Table 5.** Microbial DNA copies from in vitro rumen fermentation at 6 h, 12 h, and 24 h.

| Target Microorganism | Time (h) | Treatment [1] | | | | SEM [2] | p Value [3] | | |
| | | Korean Native Goats | | Hanwoo Steers | | | | | |
| | | RH | CF | RH | CF | | S | T | S × T |
|---|---|---|---|---|---|---|---|---|---|
| General Bacteria | 6 | 8.33 | 8.32 | 7.80 | 7.82 | 0.066 | <0.001 | 0.923 | 0.809 |
| | 12 | 8.39 | 8.49 | 7.70 | 7.87 | 0.036 | <0.001 | 0.024 | 0.443 |
| | 24 | 8.53 | 8.41 | 7.85 | 7.87 | 0.045 | <0.001 | 0.337 | 0.239 |
| Protozoa | 6 | 6.44 | 6.42 | 7.74 | 7.46 | 0.103 | <0.001 | 0.213 | 0.301 |
| | 12 | 6.21 | 6.04 | 7.11 | 7.00 | 0.077 | <0.001 | 0.227 | 0.774 |
| | 24 | 5.80 | 5.64 | 6.05 | 5.53 | 0.084 | 0.476 | 0.008 | 0.094 |
| Total Fungi | 6 | 0.81 | 1.92 | 1.50 | 1.51 | 0.339 | 0.724 | 0.188 | 0.198 |
| | 12 | 1.18 | 0.79 | 1.71 | 0.93 | 0.311 | 0.398 | 0.161 | 0.617 |
| | 24 | 0.94 | 0.89 | 1.16 | 0.90 | 0.240 | 0.683 | 0.587 | 0.724 |
| *Butyrivibrio fibrisolvens* | 6 | 7.65 | 7.55 | 7.91 | 7.89 | 0.082 | 0.009 | 0.539 | 0.689 |
| | 12 | 7.69 | 7.63 | 7.84 | 8.01 | 0.064 | 0.005 | 0.450 | 0.128 |
| | 24 | 7.88 [b] | 7.58 [c] | 8.16 [a] | 8.15 [a] | 0.053 | <0.001 | 0.039 | 0.049 |
| *Fibrobacter succinogenes* | 6 | 1.68 | 1.93 | 4.54 | 4.70 | 0.107 | <0.001 | 0.121 | 0.725 |
| | 12 | 2.95 | 4.16 | 4.66 | 5.35 | 0.126 | <0.001 | <0.001 | 0.112 |
| | 24 | 3.95 [c] | 6.09 [a] | 5.48 [b] | 6.26 [a] | 0.136 | <0.001 | <0.001 | 0.001 |
| *Ruminococcus albus* | 6 | 6.15 | 7.06 | 5.58 | 5.81 | 0.261 | 0.041 | 0.166 | 0.388 |
| | 12 | 6.84 | 6.04 | 5.74 | 5.73 | 0.195 | 0.012 | 0.101 | 0.106 |
| | 24 | 7.36 | 7.21 | 7.10 | 6.00 | 0.268 | 0.046 | 0.079 | 0.164 |
| *Ruminococcus flavifaciens* | 6 | 3.14 [c] | 3.73 [a] | 3.24 [b] | 3.20 [b] | 0.107 | 0.111 | 0.054 | 0.032 |
| | 12 | 4.64 | 4.70 | 3.15 | 3.11 | 0.079 | <0.001 | 0.866 | 0.587 |
| | 24 | 5.22 [a] | 5.05 [a] | 4.34 [b] | 3.62 [c] | 0.088 | <0.001 | 0.001 | 0.019 |

[1] Treatment: RH, rice hay; CF, cotton fiber; [2] SEM, standard error of the mean; [3] S, species; T, treatment; S × T: interaction between species and treatment. [a–c] Means with different superscripts in a row differ significantly ($p < 0.05$).

*3.3. Microbial DNA Copies from In Vitro Rumen Fermentation of Rice Hay and Cotton*

Microbial DNA copies from in vitro rumen fermentation after 24 h of incubation are shown in Table 5. No significant effect was observed in the interaction of species and treatments. However, species significantly affected bacterial abundance at 6 to 24 h of incubation, while bacterial abundance was significant across treatments at 12 h of incubation ($p = 0.024$). Quantification showed that general bacteria were higher ($p < 0.01$) in KNG than in Hanwoo steers at all incubation times. In addition, protozoa were significantly affected by species only after 6 to 12 h of incubation and by treatments after 24 h incubation. The total amount of fungi was not altered by any of the species, treatments, or their interactions.

Significant differences were observed between the species, treatment, and their interactions in microbial DNA copies of *B. fibrisolvens*, *F. succinogenes*, and *Ruminococcus flavifaciens* after 24 h of incubation. *Butyrivibrio fibrisolvens* had a higher abundance in Hanwoo steers than in Korean native goats ($p < 0.01$) and was higher in RH than in CF ($p < 0.05$). The same pattern was observed in the case of *F. succinogenes,* which had a higher abundance in Hanwoo steers than in Korean native goats ($p < 0.01$); however, the DNA copies were higher in RH than in CF ($p < 0.01$). The abundance of *R. flavifaciens* was higher in Korean native goats in both treatments than in Hanwoo steers ($p < 0.01$), but this particular species had a higher abundance ($p < 0.05$) in RH than in CF. Similarly, *R. albus* DNA copies were higher ($p < 0.05$) in Korean native goats than in Hanwoo steers. The variation in ruminal fermentation characteristics, in vitro DM degradation, and rumen microbiome of Hanwoo steers and Korean native goats were remarkable based on the data obtained using the two different fiber substrates, rice hay and cotton fiber. The rumen microbiota varies between contrasting species of ruminants, which affects rumen fermentation parameters and the degradability of the substrate.

## 4. Discussion

The rumen has a great diversity of prokaryotic and eukaryotic micro-organisms that allow the ruminant to utilize lignocellulose material to obtain energy [29]. The type of forage affects feed colonization by rumen microbes and subsequent digestion [30]. The abundance of some bacteria and archaea taxa was influenced by the host's genotype and host species influencing the rumen digestive function [31,32]. An increased understanding of this complex microbiome, the feed factors that affect rumen microecology, and the influence of the host on the rumen function should allow us to develop approaches that maximize the utilization of fibrous materials. This study compared the in vitro rumen fermentation of Korean native goats and Hanwoo steers with different fiber materials as rice hay (RH) and cotton fiber (CF) and showed an interesting difference between the species, fiber sources, and their interactions in fermentation, dry matter (DM) degradation, and ruminal microbial communities.

In this study, $NH_3$-N production was higher in Korean native goats than in Hanwoo steers. This result is in accordance with the findings of Toral et al. [33], who reported that $NH_3$-N concentration was approximately twice as high in goats than in cows. A consistent pattern was also observed in Korean native goats, with higher concentrations of acetate, butyrate, and total VFAs produced at 6 to 24 h of incubation than in Hanwoo steers. In addition, propionate had the highest concentration after 6–12 h and the acetate to propionate ratio only during 12–24 h of incubation. Furthermore, acetate, butyrate, and acetate-to-propionate ratios were higher in RH in both ruminant species after 24 h of incubation. Our findings are in agreement with the results of Toral et al. [33], who found that the molar proportions of propionate and butyrate were significantly higher in caprine species than in bovine species.

The in vitro DM degradation was higher ($p < 0.01$) in RH than CF in all of the incubation times in both species and higher ($p < 0.01$) in Hanwoo steers than in Korean native goats at 24 h of incubation than in other incubation time. Tafaj et al. [5] reported that low-fiber diets provide better rumen conditions for fiber digestion than diets containing high fiber, which supports the findings of the present study. The crude fiber and ADF

content of CF (84.88 and 85.33%, respectively) were twice as high as the RH (40.80 and 44.98%, respectively). The almost twice as high ADF in CF compared to RH is a major factor that could be adduced to lower digestibility. Moreover, ADF negatively correlates with digestibility. Additionally, in situ disappearance rate of NDF was greater for alfalfa hay and grass hay than for low-quality lovegrass hay and the extent of DM disappearance (96 h) was greater for steers than wether [34]. The order of decreasing abilities of cattle, sheep, goat, and deer to digest fiber is the opposite of their respective abilities for selective feeding [35]. Hofmann [35] noted that cellulolytic activity in the rumen of selectors is lower than in other feeders, and smaller animals in any class are usually less able to digest forage. Playne [36] suggested that greater digestion of forages by cattle compared with sheep might result in part from greater recycling of nutrients to the rumen. However, apparent total tract digestibility for NDF and ADF with ruminant species × diet was not affected by ruminant species when consuming alfalfa hay or grass hay [34]. The in situ and in vitro technique estimates only the ability of the rumen microflora to degrade forages and does not account for differences in rumination, mastication, rate of passage, or other physical factors that would influence digestion in vivo. Therefore, in the future, various studies should be conducted according to the purpose of the experiment with in vitro, in situ, and in vivo tests, selectively. However, the results of our study now show that the in vitro digestibility was better by rumen microbial action of Hanwoo steers than Korean native goats.

The feed types affect the rumen bacterial diversity, and the shaping of the rumen microbiome depends on various factors, such as the anatomical and physiological adaptation of different species, together with feed type and feeding system from various developmental stages [37]. Henderson et al. [38] noted that the differences in microbial community compositions were predominantly attributable to diet. However, Lee et al. [39] reported that the bacterial community of bovine breeds stays unique regardless of the diet, suggesting that an individual animal has own distinct bacterial community. In addition, the microbial concentrations in the rumen vary over time with respect to feeding and the position of the rumen [27]. For the microbial quantification, we found that general bacteria were higher in Korean native goats than in Hanwoo steers at all incubation times. *Butyrivibrio fibrisolvens* had a higher abundance in both RH and CF substrates in Hanwoo steers than in Korean native goats. Meanwhile, *F. succinogenes* had a higher abundance in Hanwoo steers than in Korean native goats and was higher in CF. *R. flavifaciens* had a higher abundance in Korean native goats than in Hanwoo steers and higher in RH of both species. In addition, *R. albus* was higher in Korean native goats than in Hanwoo steers. Similarly, Moon et al. [31] reported that bacteria and fungi in the rumen fluid were 2.6 times greater in goats than in cattle. The genus *Butyrivibrio* isolates accounted for 10% to 30% of total culturable bacteria in the rumen [40,41]. Zhu et al. [42] reported that the *Butyrivibrio* group bacteria represented 12.98% of the total bacteria in the rumen of goats. *F. succinogenes* is one of the dominant cellulolytic bacteria in the rumen [28]. A previous report by Lee et al. [43] noted that *F. succinogenes* had higher population in high forage diet compared to a low fiber diet. The rumen of black Korean native goat contained a very low proportion of *Fibrobacteres* (<0.1%), which was among the predominant phyla in the other rumen, Hanwoo steers and Holstein-Friesian dairy cattle (2.8 to 4.8%) [39]. *Bacteroidetes* and *Firmicutes* were the most predominant microbial phyla, and *Ruminococcus* and *Lachnospiracea incertaesedis* were in high abundances as fiber-digesting bacteria in the rumen of goats [44]. *Ruminococcus* is composed of two strong fiber-digesting bacteria species, *Ruminococcus albus* and *Ruminococcus flavefaciens*, which can produce a large number of cellulases and hemicellulases [45]. This supports the findings of the present study, the high digestibility of RH. Currently, this study demonstrated the comparison of in vitro rumen fermentation, DM degradation, and rumen microbial community of Korean native goats and Hanwoo steers using the fiber substrates RH and CF. However, a limitation of the study is the lack of control, which, unfortunately, was not measured.

Overall, the variation in ruminal fermentation characteristics, in vitro DM degradation, and rumen microbiome of Hanwoo steers and Korean native goats was remarkable based on the data obtained using the two different fiber substrates, RH and CF. Furthermore, the rumen microbiota varies between different species of ruminants, which affects rumen fermentation and the degradability of the substrate. Thus, future studies focusing on host-specific bacteria and substrate-specific bacterial action for ruminal fiber digestion to improve ruminal functions of forage utilization between suitable ruminant species should be conducted.

**Author Contributions:** Conceptualization, H.-G.S.; data curation, S.-H.K.; formal analysis, S.-H.K.; methodology, H.-G.S.; software, S.-H.K.; validation, H.-G.S. and S.-H.K.; investigation, H.-G.S. and S.-H.K.; writing—original draft, S.-H.K. and H.-G.S.; writing—review and editing, H.-G.S. and S.-H.K. All authors have read and agreed to the published version of the manuscript.

**Funding:** This research was funded by a grant from the National Research Foundation of Korea (NRF) (grant number: NRF-2020R1F1A1076625), funded by the Korean government (MSIT).

**Institutional Review Board Statement:** All experimental procedures were performed in accordance with the Animal Experimental Guidelines of the Sunchon National University Institutional Animal Care and Use Committee (SCNU-IACUC), Republic of Korea. The experimental protocols were reviewed and approved by SCNU-IACUC (approval number: SCNU IACUC-2018-01).

**Informed Consent Statement:** Not applicable.

**Data Availability Statement:** Upon reasonable request, the datasets of this study can be available from the corresponding author.

**Conflicts of Interest:** The authors declare no conflict of interest.

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
