# Peer review of "Effects of Different Fiber Substrates on In Vitro Rumen Fermentation Characteristics and Rumen Microbial Community in Korean Native Goats and Hanwoo Steers"

_fermentation, doi:10.3390/fermentation8110611_

Round 1

Reviewer 1 Report

Kim et al investigated the rumen microbiome of the Korean native goats and Hanwoo cattles and found out distinct differences in microbiome composition between the species, as well as digestibility patterns in general and upon intoducing different feed type. 

I would recommend the authors to consider the following points:

1. Were the anaerobic conditions ensured during the transfer of the rumen fluid into the anaerobic in vitro system? If not, the authors should consider stating this as a limitation of the study, as some of the strictly anaerobic bacteria might have been lost. 

2. Authors should consider specifying what "general bacteria" means? (e.g. page 6, line 9).

3. Authors should consider being more consistent when naming the animal species they investigated (e.g. sometimes they are named as Hanwoo steers, sometimes as Hanwoo cattles, etc.)

4. Authors should consider a good proof-reading of the article to avoid typos (e.g. additional bracket and dot page 1, line 4 of the abstract).

Author Response

We appreciate the reviewer for their precious time in reviewing our paper and providing insightful comments. 

Reviewer 2 Report

The manuscript could be improved after major revisions. comments are listed below.

Major points:

1. In general, the study is not strictly controlled. Rumen is a complex environment. Without controlling protein and energy, we can't just say that fiber affects rumen fermentation. In addition, there are many fiber degrading bacteria, and simple quantification of these four bacteria cannot fully explain rumen functions. The discussion part is also slightly insufficient, and the statement of difference in the table is not clear enough. By the way, where are the line numbers?

2. There is a problem with the storage method. It is necessary to introduce CO2 to maintain anaerobic conditions, please explain.

3. Lack of control in the experiment, please explain.

Minor points:

1. Title

Title could be refined since "depend on fiber substrates" is not proper here, in addition, in vitro rumen fermentation should be emphasized

2. Abstract

"and cotton fiber (CF). )." remove "). ";

"(p < 0.05)" un-unified format;

"Overall results suggest " suggested.

3. Material and method

Please add experiment site;

"(73 ± 12 kg body weight; " Lack of ")";

please add Analysis method of chemical composition of diet.

4. Results

"volatile fatty acids (VFA) " should be VFAs

 "Consequently, propionate had the highest concentration (p < 0.05) only after 6–12 h and the acetate to propionate ratio only during 12–24 h of incubation. Acetate, butyrate, and acetateto-propionate ratios were higher (p < 0.05) in RH than in CF for both ruminant species after 24 h of incubation. "This result statement is inconsistent with the table information;

"protozoa were significantly affected by species only after 6 to 14 h of incubation" 14h?

5. Discussion

"In this study, NH3-N production was higher in Korean native goats than in Hanwoo steers." please explain the reason.

"Butyrivibrio fibrisolvens had a higher abundance in Hanwoo steers than in Korean native goats and was higher in RH. " Not clear

Author Response

(The authors gave the same response as above.)

Reviewer 3 Report

General comment

The study is an interesting one. However there are methodological concers that need to be addressed. The discussion reflects a good review of recent literature to validate the results obtained. A more lucid conclusion/recommendation at the end of the discussion is necessary. English language proof reading is essentially required as there are errors across several sections of the manuscript. Without line numbering, it has been difficult to highlight all of them.Pre-washing of substrate before a second incubation appears unorthodox. Authors need to justify this. Specific comments are detailed below:

Abstract

Line 3: Sentences need correction.

Methodology

2 weeks. Write in full

Were the filtered rumen fluid from the 4 animals from each specie pooled together? What constitute the experimental units? Replicates for each specie across time interval of 6, 12 & 24h? This is not clear in the procedures. How many times was each incubation/specie carried out? Stepwise documentation of protocol makes repeatability of a study possible. Authors should implement this.

Water at 39-40°C cannot be hot water, can it? Please correct which is wrong.

What was the purpose of the second incubation after immersion in 39-40C water? DM degradation? Bearing in mind that the pre-washing removes very soluble nutrients and therefore, one can expect a significant longer lag time in rumen activity. This needs to be justified.

The RH and CF of the chemical composition….the sentence is not clear.

Section 2.3

Samples of fermentation…this should be rumen fluid after incubation? Filterate from incubation bottles?

Statistical analysis:

Authors should include the statistical model used in the analysis. What were the sources of error considered when comparing treatments?

The standard unit for molar proportion of VFAs is mol/100mol. Authors should re-calculate the molar proportions of each VFA and report the proportion relative to 100. This may require a re-run of the SAS analysis as the values will expectedly change.

Table 3: The statistical procedures use in comparing treatments (RH vs CF) and across two livestock species (KNG vs Hanwoo) is faulty. A statistician’s expertise might be required. For example, when there are no significant interaction, simple effects of factors as treatments are reported (either dietary treatment or specie). Assigning abc values where this is not so, is misinterpretation of result.

Table 4: How was the digestibility study carried out? DM digestibility of CF in Hanwoo seems to be 3.29% at 6h and 1.18% at 12 h and then 15.93% at 24h. This is not a plausible result, except there is considerable error in the results reported.

The DM digestibility’s of the treatments are too low and this is associated with the the substrates themselves and the method of analysis. The In vitro Dry matter digestibility procedure of “Tilley and terry method as modified by Engels and Van der Merwe” recommends the addition of urea during incubation which improves DM digestibility of substrates with very low crude protein content.

Section 3.3

The authors reported as follows “Microbial DNA copies from in vitro rumen fermentation after 24 h of incubation are shown in Table 5. No significant differences in the abundance of general bacteria were observed between treatments and in the interaction of species and treatments; however, species significantly affected bacterial abundance after 6 to 24 h of incubation.” However, the table shows that total bacteria at 12h was significant across treatments (P = 0.024).

Discussion

Aside from language errors, the discussion section looks ok. What is the implication of differing microbiomes across the goat and cattle on their feed utilisation potential within the farming system where these animals are reared? This should cap the discussion as it provides a form of recommendation to end users of this research.

Author Response

(The authors gave the same response as above.)

Round 2

Reviewer 2 Report

The manuscript has been improved and I would like to suggest the author to discuss the limitation of this study (lack of control) in the end of discussion part. Thanks.

Author Response

The manuscript has been improved and I would like to suggest the author to discuss the limitation of this study (lack of control) in the end of discussion part. Thanks.

            Thank you for your valuable comments and suggestions. We included a statement regarding the limitation of the study (Lines 319-322).

“Currently, this study demonstrated the comparison of rumen fermentation, DM degradation, and rumen microbial community of Korean native goats and Hanwoo steers using the fiber substrates RH and CF. However, a limitation of the study is the lack of control, which, unfortunately, was not measured.”

Reviewer 3 Report

The authors have corrected the manuscript based on previous recommendations. However, a major issue remains with the statistical analysis for which corrections are required. Ascribe superscripts only when interaction effects are significant (see table 5). Include statistical model statement.

Other minor corrections are detailed below: 

L14: Correct sentence “…consuming rice hay (RH) and cotton fiber.

L14-15: Ph, ammonia and total VFA higher in what compared to what? It has to be in relation to something.

L62-66: These statements are results or conclusions of study. Remove them from introduction.

The methodology has been refined considerably. How many times was the in vitro incubation procedure done? If only once, ignore but ideally, 3 replicate runs should have been carried out.

Statistical analysis

Authors may try to include a statistical model statement. This will help to clarify how the factors were ordered before comparison was made.

Table 4: Which experimental protocol was used to implement this? This is not documented in Materials and method. Was it the same in vitro gas production protocol? Authors should provide sufficient information on this.

Table 5: The authors are in error to assign superscripts to means when interaction effect of Species and treatment are not significant.

L233: R. albus DNA copies were higher….

L233-235: This is not part of your result. Move it to discussion section.

L259-261: Rumen ammonia is a function of effectively degraded nitrogen (feed CP and NPN). It may not be indicative of bacteria abundance or to microbial protein synthesis.

L270-275: The almost twice as high ADF in CF compared to RH is a major factor that could be adduced to lower digestibility. ADF negatively correlates with digestibility.

L314: rumen not rumens…black Korean native goat or black KNG.

L315: rumen not rumens.

Author Response

Thank you very much. We appreciate you for taking the time to review our manuscript and giving valuable comments and suggestions. We have revised the manuscript based on the comments and suggestions.
